# Scientometric Analysis and Classification of Research Using Convolutional Neural Networks: A Case Study in Data Science and Analytics

**Mohammad Daradkeh** [1,2,*], **Laith Abualigah** [3,4] , **Shadi Atalla** [1] and **Wathiq Mansoor** [1]

1. College of Engineering and Information Technology, University of Dubai, Dubai 14143, United Arab Emirates; satalla@ud.ac.ae (S.A.); wmansoor@ud.ac.ae (W.M.)
2. Faculty of Information Technology and Computer Science, Yarmouk University, Irbid 21163, Jordan
3. Faculty of Computer Sciences and Informatics, Amman Arab University, Amman 11953, Jordan; aligah.2020@gmail.com
4. Faculty of Information Technology, Middle East University, Amman 11831, Jordan
* Correspondence: mdaradkehc@ud.ac.ae

**Abstract:** With the increasing development of published literature, classification methods based on bibliometric information and traditional machine learning approaches encounter performance challenges related to overly coarse classifications and low accuracy. This study presents a deep learning approach for scientometric analysis and classification of scientific literature based on convolutional neural networks (CNN). Three dimensions, namely publication features, author features, and content features, were divided into explicit and implicit features to form a set of scientometric terms through explicit feature extraction and implicit feature mapping. The weighted scientometric term vectors are fitted into a CNN model to achieve dual-label classification of literature based on research content and methods. The effectiveness of the proposed model is demonstrated using an application example from the data science and analytics literature. The empirical results show that the scientometric classification model proposed in this study performs better than comparable machine learning classification methods in terms of precision, recognition, and F1-score. It also exhibits higher accuracy than deep learning classification based solely on explicit and dominant features. This study provides a methodological guide for fine-grained classification of scientific literature and a thorough investigation of its practice.

**Keywords:** scientific literature; thematic classification; scientometric; deep learning; convolutional neural network (CNN)

## 1. Introduction

In recent years, the number of new research disciplines and practices has grown manyfold, leading to an internal diversification of research and greater interdisciplinary interactions. This evolution of interdisciplinary research paradigms has led to the development of new research foci, organization of interdisciplinary conferences, establishment of international scientific societies, and creation of new journals, among others [1]. As the landscape of scientific disciplines continues to differentiate, classification systems are being developed to better reflect this dynamic reality and facilitate the study of knowledge production and dissemination, and as such can serve important classification functions [2–5]. Scientometric analysis and the classification of scientific literature provides an indispensable basis for the evaluation and synthesis of published literature and improves the efficiency of researchers' information search. At the same time, it helps academic institutions and scientific literature management platforms analyze the development direction of disciplines [6], facilitates the exploration of knowledge production and dissemination, and accelerates the rapid development of scientific research [7,8]. Acknowledging the advantages of scientometric analysis, it has been widely used to evaluate leading scientific researchers or

publications [9], examine the structure of a scientific field's network [10,11], reveal emerging issues [12], and help researchers study the development of research fields and disciplines by categorizing documents along multiple dimensions [4].

Typically, scientometric studies focus on broad classification of published articles based on primary or secondary subjects or disciplines. Currently, scientometric studies classify publications using generic classification systems, such as the Web of Science (WoS) subject categories and the Field of Science and Technology Classification (FOS) [9]. In their current form, these systems are too broad to adequately reflect the more complex, fine-grained cognitive reality; therefore, their scope is limited and they only indicate broad scientific domains or general disciplines. Empirical studies [5] as well as theoretical arguments by researchers [13] emphasize the need for fine-grained classification approaches. A recent study by Wahid et al. [11] found that focused research communities can be distinguished based on publication associations, and publication practices and patterns may vary within these communities. However, fine-grained classification is challenging for researchers because it is not clear to what extent authors in particular fields collaborate to disseminate new findings and knowledge [9,14,15]. Such finer classification usually involves two aspects. First, it narrows the disciplinary focus of a particular research area into categories; for example, the literature on data science and analytics can be further divided into big data analytics, business analytics, business intelligence, machine learning, predictive modeling, and deep learning. Second, the classification dimension includes research content and research method; for example, a research paper can be classified as business analytics based on its content and as empirical research based on its research methodology.

Recently, there has been a resurgence of interest in applying machine learning methods to scientometric classification and analysis of scientific literature, as these algorithms have achieved acceptable results in text analysis and classification. Commonly used machine learning algorithms include support vector machines (SVM), Naive Bayes classifiers, and the K-nearest neighbor model (KNN) [16]. However, with the abundance and diversity of scientific research and the exponential increase in scientific output, classification methods based on general scientometric information and traditional machine learning methods have shown significant shortcomings in terms of coarse classification and insufficient accuracy [1,9,17]. Moreover, the data elements used by existing scientific literature classification systems are primarily derived from explicit scientometric information such as titles, abstracts, and keywords. Nevertheless, data elements may also have implicit relationships such as journal names, authors, research institution names, and research content and method [6,10,18,19]. The same journal, author, or research institution are more likely to focus on certain research content and methods, even if there is no direct relationship between them. Therefore, to improve the accuracy and performance of scientific literature classification, it is imperative to study and integrate these implicit relationships along with the explicit features of scientific publications.

To this end, this study develops a deep learning approach for scientometric analysis and classification of scientific literature based on convolutional neural network (CNN), aiming at dual-label classification of literature based on research content and method. We demonstrate the efficacy of the proposed model using an application example from the data science and analytics literature. The empirical results show that the scientometric classification model proposed in this study performs better than comparable machine learning classification methods in terms of precision, recognition, and F1-score. It also exhibits higher accuracy than deep learning classification based only on explicit and dominant features. It is worth noting, though, that this study only considered publications from a single domain, namely data science and analytics. Nevertheless, we investigate the use of textual content to classify publications in three major databases and platforms (Scopus, ProQuest, and EBSCOhost). As will be discussed in more detail below, we aim to classify these data science and analytics abstracts into granular subcategories, since an article can be assigned to multiple categories simultaneously. The novelty of this study is that we applied a unique approach to validate the data collected for our machine learning experiments and

were able to assign multiple sub-disciplinary categories to a single article. The results of this study may therefore open up new opportunities to understand knowledge production and dissemination in the emerging sub-disciplines of data science and analytics at a more detailed level.

## 2. Literature Review

Scientometrics is a field of study concerned with the quantitative analysis of textual features and characteristics of scientific literature [7,8,14]. It can be considered as the science of science. The goal of scientometrics is to evaluate the development of a scientific field, influence of scientific publications, patterns of authorship, and production processes of scientific knowledge. Typically, scientometrics is concerned with monitoring research, evaluating the scientific contribution of authors, journals, and specific works, and assessing the dissemination of scientific knowledge [20]. As part of these approaches, researchers develop methodological principles for deriving information from communication activities and use specific methods to achieve these goals, including citation analysis, social network analysis, syndicated terminology analysis, and text mining [16]. Scientometric studies usually focus on authorship or measurement of journal or professional association contributions. However, they may also examine terms that appear in titles, abstracts, full texts of book chapters and journal articles, or keywords assigned by editors to published articles or publishing houses [9,21–23]. González-Alcaide et al. [24] used scientometric analysis to identify the main research interests and directions on Chagas cardiomyopathy in the MEDLINE database. Specifically, they identify research patterns and trends on Chagas cardiomyopathy. Similarly, Mosallaie et al. [12] used scientometric analysis approaches to identify trends in artificial intelligence in cancer research, while Wahid et al. [11] applied scientometric analysis and group-level comparative analysis of Pakistani authors to determine their scientific productivity. The body of scientific knowledge in a particular area of interest provides a comprehensive description and representation of previous and current knowledge on that particular topic. Therefore, a comprehensive and quantitative analysis of such literature sources provides valuable insight into the evolving research interest in the field and can provide a comprehensive picture of the topics and their current status and relevance.

Scientometric analysis is based on the breakdown and disclosure of relationships between different research contributions (articles), from which statistical indices are calculated to reveal research paradigms and emerging trends [13]. In scientometric studies, scientific literature is often classified using abstracts, keywords, and titles. For example, Hernandez-Alvarez et al. [23] used high-frequency keywords in abstracts, manually identified important terms to generate knowledge domains, and divided the literature into multiple knowledge domains by calculating the similarity between them to classify the literature. Similarly, Kim et al. [25] used keywords and abstracts as raw data to create a document terminology matrix representing the frequency of terms in the accounting literature. Makabate et al. [3] used a grounded theory approach to code the terms collected in the abstracts and classify research methods. Nosratabadi et al. [26] extracted feature terms from titles and abstracts based on lexicality, filtered them, calculated feature term frequencies to characterize the documents as feature vectors, and classified them using association rules. Ozcan et al. [10] constructed a literature topic matrix and a topic-feature term matrix based on the titles and abstracts of patent documents to identify topics for technological innovations. In addition, several researchers have attempted to construct features from external resources to improve classification accuracy. For example, Purnomo et al. [27] used external feature information such as Wikipedia and news sites to improve the accuracy of literature classification. Bhatt et al. [28] used Medical Subject Headings (MeSH) as the basis for selecting key terms related to stem cells as feature vectors to characterize the literature. Ho and Shekofteh [29] selected patent classification codes to create a technical lexicon. They characterized the literature as vectors based on the Derwent Manual Code (DMC) to create a patent manual code matrix and build a technical knowledge map of the patent literature.

In recent years, machine learning methods have been widely used to improve the accuracy of scientometric classification and analysis of scientific literature, as these algorithms have achieved acceptable results in text classification. For example, Eykens et al. [9] used supervised machine learning methods (gradient boosting and Naive polynomial Bayes) to classify social science articles based on textual data. Similarly, Huang et al. [2] improved the NB algorithm by using local weighting to improve classification performance. Makabate et al. [3] compared the performance of KNN and SVM algorithms in classifying scientific literature in data analytics. As deep learning algorithms have evolved, convolutional neural networks (CNNs) have been shown to automatically learn features from text and reduce manual input of feature information, resulting in better text classification capabilities than traditional machine learning algorithms. In their study, Salazar-Reyna et al. [30] applied different machine learning algorithms to classify documents and found that deep learning algorithms are superior to traditional machine learning algorithms. Sood et al. [4] performed a multi-level classification of 1.7 million documents information in an international scientific journal index based on CNN models and obtained satisfactory scientometric classification results.

Machine learning methods using neural networks and BERT (Transformers Bidirectional Encoder Representations) models have also been used separately to vectorize and reveal relationships between scholarly articles. Kandimalla et al. [31] presented a comprehensive characterization study using neural networks and word embedding models to organize articles by WoS topic categories and focused on the use of these moderately novel NLP strategies to vectorize logical distributions. The researchers show that such frameworks perform excellently in grouping samples, achieving a normal F-score of 0.76, with values ranging from 0.5 to 0.95 for each topic category. As a result, subcategories with too many record pairs are either grouped or removed from the current study because they negatively affect classification performance. In addition, documents with more than one classification continue to be excluded. The authors conclude that their analysis shows that the managed learning approach scales better than strategies based on reference clusters. Dunham et al. [32] train SciBERT classifiers on arXiv metadata, which are then used to develop a model for distinguishing the important artificial intelligence distributions in WoS, digital science dimensions, and Microsoft academic. The authors report that F1 values range from 0.58 to 0.85 for the four classifications in the artificial intelligence domain.

Machine learning and deep learning algorithms have been used in scientometric research mainly based on explicit scientometric information and features such as abstracts, keywords, and titles. However, in addition to these explicit features embedded in the literature, scientometric information also includes journal name, author name, and institution name. These relevant contents, however, are still scarcely investigated. Wang et al. [18] found that the classification accuracy of data elements such as journal title, author, and institution added directly to the feature vector does not improve regardless of whether traditional machine learning or deep learning algorithms are used, but decreases significantly. According to their findings, this is mainly due to the fact that the technical terms referring to research content and method can be obtained from abstracts, keywords, and titles. At the same time, there are almost no technical terms in the journal, author, and institution names that can directly characterize literature information. Each journal defines its research focus and emphasizes specific research methods, each author has their research expertise and focuses on research methods, and each research institution or team also develops specific research areas and common research methods. Throughout this study, we conjecture that these data have implicit features that relate to research content and research method. Therefore, our primary goal is to use the implicit textual features of scientific publications to classify them into pre-established discipline-specific categories. As we elaborate below, we intend to classify these scientific literature documents into granular subcategories, implying that a scientific publication can be simultaneously assigned to one or more categories.

## 3. Model Development

To classify scientific literature, scientometric information is used as a basis, an initial feature matrix is created by feature extraction, and terms vectorization is performed. In this study, scientometric data are divided into explicit and implicit features, where abstract, keywords, and title are explicit features, while journal title, authors, and institutions are implicit features. Explicit and implicit features differ significantly in their identification methods; explicit features can be easily identified by their direct association with the classification label [33]. In contrast, there are no obvious linguistic and syntactic clues for identifying implicit features; instead, they must be determined based on their deep semantic features and are usually those features that are not directly associated with the classification label [34]. Therefore, identifying implicit features is still one of the most difficult tasks in scientometric analysis and classification of literature. A very effective and useful method to extract implicit features is to use association rules to identify the implicit feature by finding the association of a certain feature with the classification labels; in this study, the classification labels are presented as research content and research method. For the explicit features, the feature terms are extracted directly from the literature documents. For the implicit features, feature mapping is performed to make them explicit. In this way, a vector of feature terms is created and applied to Word2Vec to build a term vector model that feeds the CNN deep learning model. The CNN deep learning algorithm is then used for feature extraction and fine-grained classification, i.e., dual-label classification for research content and method. The output layer of the CNN model implements dual-label classification of research content and research method simultaneously.

The research methodology used in this study is detailed in Figure 1. This process is divided into three main phases: (1) creation of scientometric features and stopword lexicons. This involves creating the scientometric feature lexicon and stopword list by selecting titles, abstracts, and keywords of all documents in the training set to improve the accuracy of the features; (2) creation of the feature matrix and vectorization. This involves dividing the scientometric information into explicit features (abstract, title, and keywords) and implicit features (author, journal name, and institution). For the explicit features, term tokenization and stop word removal are performed; for the implicit features, feature mapping is performed to make the implicit features explicit. A feature word vector based on scientometric information is created and used as input data for the CNN classification model; and (3) literature analysis and classification using CNNs, where the form research content (C) × research method (M) is developed in the output layer of the CNN model to achieve dual-label classification of the literature.

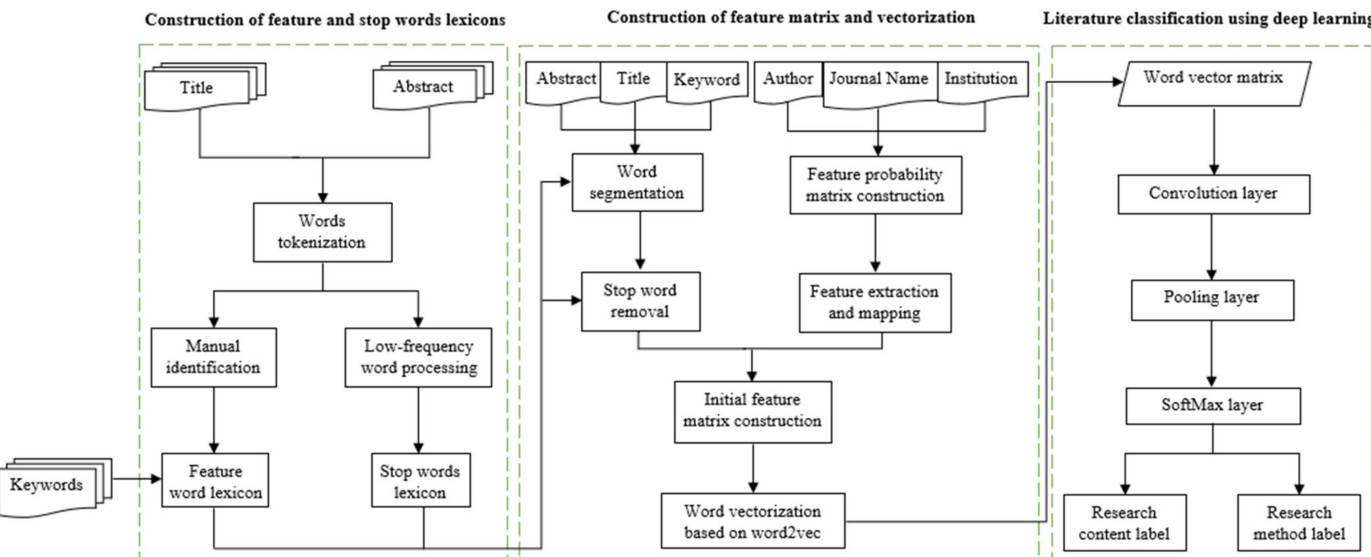

**Figure 1.** Scientometric analysis and classification of scientific literature using CNN.

In machine learning applications, dual-label classification involves assigning two target labels to each document instance. Dual-label classification is a variant of the classification problem that requires assigning two target labels to each document. Deep learning neural networks are an example of an algorithm that inherently supports dual-label classification problems. As such, they have been widely used in recent years, particularly in text analysis and opining mining, e.g., citation annotation [35], text categorization [21], movie genre categorization [36], and web mining [37]. These applications typically involve a significant number of labels, and the number of labels continues to increase with new applications. Therefore, describing samples with only a single label is challenging [38]. In this study, we address the problem of dual-label classification in the context of scientometric classification of literature, where labels for research content and method must be assigned to each document with high descriptive accuracy. The task of dual-label classification is to train a function to predict the unknown sample and return two target labels for each of the documents. In contrast to traditional classification, dual-label classification represents a sample by an eigenvector and two target labels, instead of just one exclusive label. The traditional classification approach consists of training different classification prediction labels separately. This approach is characterized by low training and testing efficiency and reasonable memory consumption when the set of target labels is quite large [36,39,40].

### 3.1. Building Stop and Feature Terms Lexicons

Building feature and stop terms lexicons is an integral part of data preprocessing. The lexicon of feature terms is used as a custom dictionary to improve the accuracy of automatic term tokenization. The lexicon of stop terms helps filter out the noise in the term tokenization results, improving the deep learning model's performance and avoiding the phenomenon of overfitting. This study builds a feature lexicon and a stop term lexicon for the literature domain based on all the data elements of title, abstract, and key terms in the training set. The feature term lexicon consists of three main parts. First, given the importance of key words in scientometric information, all key words were included in the feature term lexicon. Second, the feature term lexicon included high-frequency terms (greater than or equal to 5) included in titles and abstracts. Finally, typical terms describing the literature topics were included in the feature term lexicon in coordination with the knowledge of subject matter experts. Validation of terms for inclusion in the final lexicon was performed using two methods: validation by three subject matter experts and comparison with existing terms from the data science and analytics literature [4,30,41,42]. The final validated lexicon of feature terms will be used in downstream literature classification tasks by helping to select appropriate data science terms from publication platforms/databases (e.g., ProQuest, EBSCOhost, and Scopus) and pre-annotating these terms to support the development of a deep-learning application for classifying mentions of data science and analytics in the literature. Further explanation of the derived terms in the final lexicon can be found in Section 4.1.

In this study, only the English Snowball list [43] was initially used for the stop word lexicon, but the results were not satisfactory. After analysis, we found two main reasons for this situation. First, there are more formal descriptive terms in the scientific literature, such as "together with", "pointed out", "in agreement with", and other sentence-initial terms, which can easily mislead the machine learning procedure. Second, terms with low frequency (less than 5) and insignificant categorical features are prone to overfitting in machine learning. Therefore, in addition to Porter's English snowball list, the stopword lexicon in this study includes sentence-initial terms and terms with low frequency and unclear categorical features. Compared to Porter's English snowball list [43], our stopword lexicon consists of frequently mentioned terms related to open source tools (e.g., KNIME, RapidMiner, Weka, and MS Power BI), programming languages, and libraries (e.g., Python, R Script, STATA, SQL, and NLP).

### 3.2. Feature Matrix Construction and Vectorization

In text mining applications, deep learning models can automatically find features from distributed term vectors, which are highly portable and efficient to learn compared to conventional machine learning algorithms such as support vector machines (SVM) and conditional random fields (CRF) [44,45]. However, the quality of initial features still affects the efficiency of deep learning, and low-quality features tend to be overfitted or underfitted [46]. In this study, the data elements in scientometric information are divided into explicit and implicit features and processed separately.

#### 3.2.1. Extraction of Explicit Features

First, key terms are directly added to the key term set $K$. A lexicon of feature terms and stop terms is then added, and the title and abstract are segmented with a tokenization tool to form the title feature set $T$ and abstract feature set $S$, respectively. This process is shown in Equations (1)–(3) as follows:

$$K = (k_1, k_2, \ldots, k_r) \tag{1}$$

$$T = (t_1, t_2, \cdots, t_P) \tag{2}$$

$$S = (s_1, s_2, \cdots, s_q) \tag{3}$$

where $k_r$ is the $r_{th}$ term in the key terms, $t_P$ is the $p_{th}$ term in the title, and $s_q$ is the $q_{th}$ term in the abstract. Note that $r$, $p$, and $q$ are variables for each document. To determine the length of the subsequent term vectors, three hyperparameters, $R\ (\geq r)$, $P\ (\geq p)$, and $Q\ (\geq q)$, are needed to determine the length of $K$, $T$, and $S$, respectively. The incomplete part is filled with 0, as shown in Equations (4)–(6).

$$K = (k_1, k_2, \ldots, k_r, \overbrace{0, \ldots, 0}^{R-r}) \tag{4}$$

$$T = (t_1, t_2, \cdots, t_P, \overbrace{0, \ldots, 0}^{P-p}) \tag{5}$$

$$S = (s_1, s_2, \cdots, s_q, \overbrace{0, \ldots, 0}^{Q-q}) \tag{6}$$

#### 3.2.2. Implicit Feature Mapping

The institution name, journal name, and authors in the scientometric information are implicitly related to the research content or research methods of the literature. In the field of data science and analytics, academic articles published by computer science schools, for example, may focus on "applications of data analytics," while articles published by business schools may focus on "business analytics and decision making". Scholarly articles published in data analytics journals typically focus on data analytics applications and techniques, while scholarly articles published in healthcare journals typically focus on the application of data analytics in healthcare. In addition, collaboration with other researchers may change the research content or methodology. Similarly, an author usually uses a relatively specific type of research method and focuses on a particular research area, but when the author collaborates with other authors, the research content or method may change. Therefore, in this study, the scientometric information implicit in the names of the institution, journal, and authors is made explicit using feature mapping and then added to the original feature matrix. The mapping procedure for author, journal, and institution features is presented below:

(1) Author feature processing. The authors are associated with the domain literature, and the implicit feature of authorship is made explicit according to the co-occurrence frequency of the authors with the research content and research method involved in the

published literature [47]. Figure 2 shows the process of generating research method labels for different types of authors.

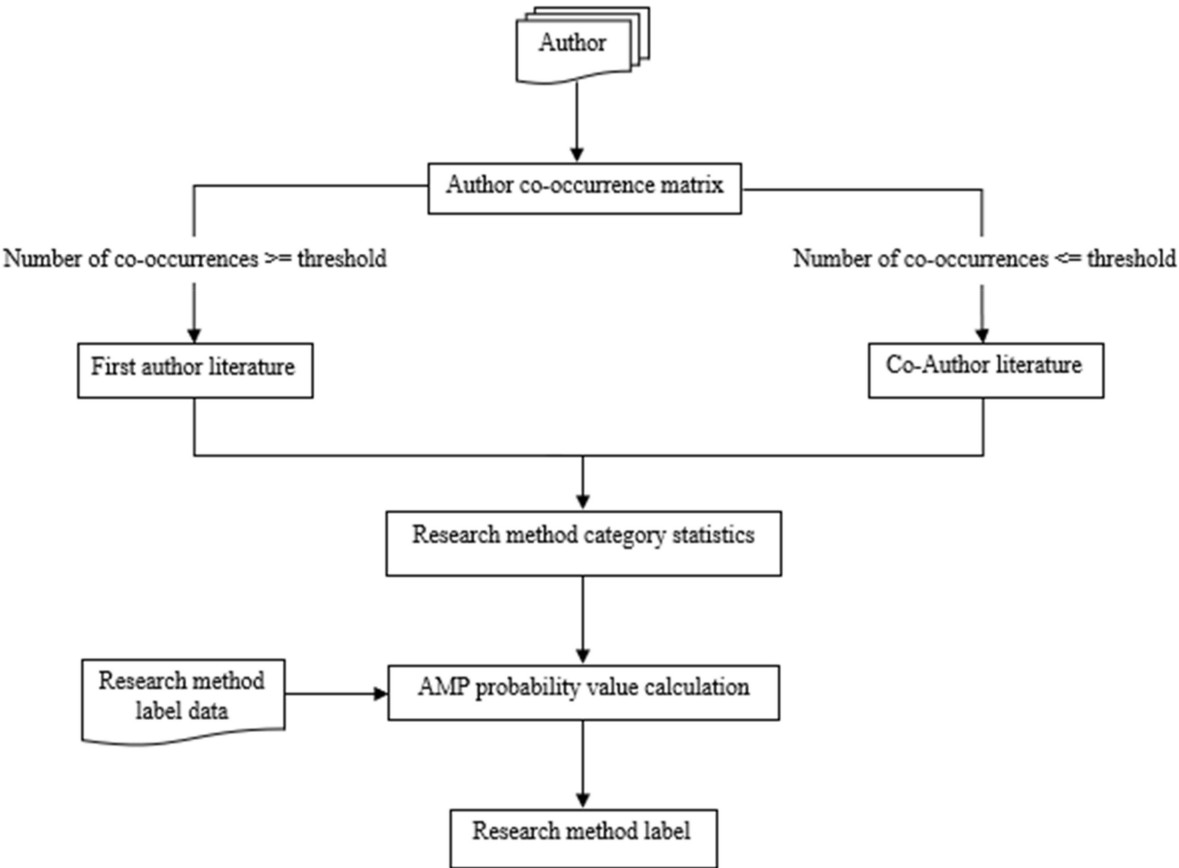

**Figure 2.** Author features mapping process.

The author co-occurrence matrix is constructed based on the author co-authorship relationship of domain literature. If the number of co-occurrences (i.e., co-authorship frequency) exceeds a specified threshold, the two authors are considered to have a stable collaborative relationship in a particular area of academic research and are considered co-authors for feature mapping; otherwise, only the first author is considered for feature mapping. Specifically, the frequency distribution table of "author–research method" is generated by first counting the frequency of co-authors (or first authors) according to the research method category. The probability value of the research method used by different authors is then calculated as *AMP*, shown in Equation (7). *AMP* indicates the percentage of papers authored by a particular author using a particular research method. For example, if author-1 publishes 10 papers, of which 5 papers use a mixed methods approach, then the *AMP* is 0.50 (50%). The larger the *AMP* value, the stronger the preference of an author for a certain research method. Finally, the "author–research method" probability distribution table is generated.

$$AMP_i(a) = \frac{m_{ia}}{\sum_{i=1}^{M} m_{ia}} \tag{7}$$

In Equation (7), $M$ denotes the category of research method in the paper and $m_{ia}$ denotes the frequency of the $i_{th}$ research method used by author $a$. Based on the probability distribution table, authors are assigned to the explicit features of the research method. First, the threshold value for the *AMP* transformation probability is set, then the label for the research content with the largest *AMP* value that satisfies the threshold value is selected, and the authors are assigned to this label. The threshold value is a hyperparameter, and after experimentation, it is assumed that the threshold value is set to 0.7. That is, if an

author's *AMP* value is greater than 0.7, the author is assigned to that research method label; otherwise, the placeholder 0 is used. Table 1 shows an example of the assignment of authors to research method labels. Author-1 is mapped to research method-2, co-author-2 is mapped to research method-1, author-5 is mapped to research method-3, while author-3 and co-author-4 are replaced by placeholder 0.

**Table 1.** Example of author–research method probability distribution.

|  | Research Method-1 | Research Method-2 | Research Method-3 | Research Method-4 |
|---|---|---|---|---|
| Author-1 | 0 | 1 | 0 | 0 |
| Co-Author-2 | 0.8 | 0.2 | 0 | 0 |
| Author-3 | 0 | 0 | 0.5 | 0.5 |
| Co-Author-4 | 0.6 | 0 | 0.4 | 0 |
| Author-5 | 0 | 0 | 1 | 0 |

(2) Journal feature processing. Similar to the author feature processing, the journal name is associated with the domain literature and mapped to the research content and research method explicit features. Taking the research content as an example, the processing flow is shown in Figure 3.

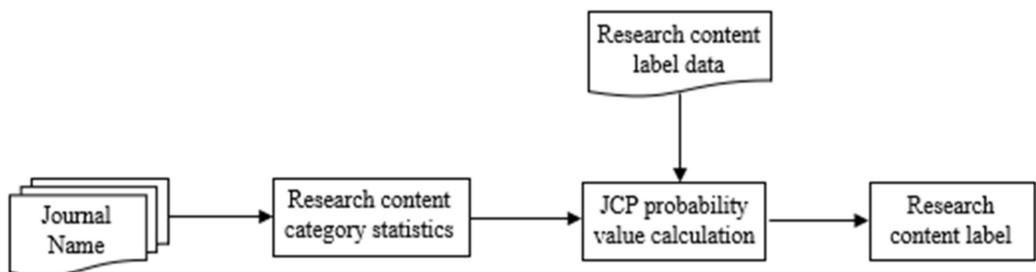

**Figure 3.** Journal features mapping process.

First, the frequency of different research contents of each journal was calculated using the journal title as the object to generate a frequency distribution table of journal–research content. The probability value of research content label for each journal was calculated as *JCP*, as shown in Equation (8). Again, the higher the value of *JCP*, the stronger the preference of a journal for a specific research content. A "journal–research content" probability distribution table is then generated based on *JCP*.

$$JCP_i(j) = \frac{C_{ij}}{\sum_{i=1}^{C} C_{ij}} \tag{8}$$

In Equation (8), *C* represents the category of the research content in the literature and $C_{ij}$ represents the frequency of journal *j* for the $i_{th}$ label of the research content. Based on the probability distribution table, the journal name is mapped to the explicit features of the research content. First, the *JCP* transformation probability threshold is set, and then the journal name is transformed to research content labels greater than or equal to the threshold. If there are no labels that meet the threshold or the number of labels is insufficient, a placeholder 0 is used instead. Assuming that the threshold is set to 0.33, labels for research content are added to the set of journal title mappings if the *JCP* value of research content is greater than 0.33. Table 2 shows examples of journal titles mapped to research content labels. The mapping set for journal-1 is (research content-1, research content-4, 0), the mapping set for journal-3 is (research content-3, research content-4, research content-5), and the mapping set for journal-5 is (0, 0, 0).

**Table 2.** Example of a journal–research content probability distribution.

| | Research Content-1 | Research Content-2 | Research Content-3 | Research Content-4 | Research Content-5 |
|---|---|---|---|---|---|
| Journal-1 | 0.60 | 0 | 0 | 0.40 | 0 |
| Journal-2 | 0.15 | 0.10 | 0.75 | 0 | 0 |
| Journal-3 | 0 | 0 | 0.34 | 0.33 | 0.33 |
| Journal-4 | 0 | 1 | 0 | 0 | 0 |
| Journal-5 | 0.25 | 0 | 0.20 | 0.25 | 0.30 |

(3) Research institution features mapping. The research institution is linked to the published article and mapped to the research content and research method explicit features. For instance, the processing flow of mapping the institution to the research content is shown in Figure 4.

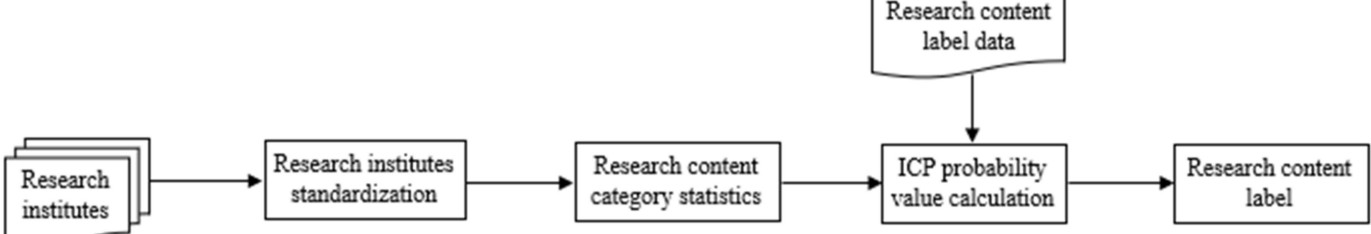

**Figure 4.** Features mapping process for research institutions.

The research institutions were first treated as follows: (i) if more than one research institution exists in the literature, only the first research institution is selected; (ii) regular expressions perform the classification of primary and secondary research institutions. Considering that the first-level research institution such as "XXXX University" basically cannot indicate the research content of the domain literature, only second-level research institutions, such as the Faculty of Economics and Management and the Faculty of Engineering, were retained for feature mapping. After that, the probability values of "research content" and "research method" labels were calculated for each research institution, and feature mapping was performed in the same way as journal mapping.

First, the frequency of the different research content for each institution name was calculated to create a frequency distribution table of institution–research content. The probability value of the research content label for each institution was calculated as *ICP*, as shown in equation (9). The higher the value of *ICP*, the stronger an institution's preference for a particular research content. Then, based on *ICP*, a probability distribution table of "institution–research content" is constructed.

$$ICP_i(j) = \frac{C_{ij}}{\sum_{i=1}^{C} C_{ij}} \tag{9}$$

In Equation (9), *C* represents the category of the research content in the literature and $C_{ij}$ represents the frequency of institution *j* for the $i_{th}$ label of the research content. Based on the probability distribution table, the institution name is mapped to the explicit features of research content. First, the *ICP* transformation probability threshold is set, and then the institution name is transformed to research content labels greater than or equal to the threshold. If there are no labels that meet the threshold or the number of labels is insufficient, a placeholder 0 is used instead. Assuming that the threshold is set to 0.33, labels for research content are added to the set of institution name mappings if the *ICP* value of research content is greater than 0.33. Table 2 shows examples of institution names mapped to research content labels. The mapping set for institution-1 is (research content-1,

research content-4, 0), the mapping set for institution-3 is (research content-3, research content-4, research content-5), and the mapping set for institution-5 is (0, 0, 0).

### 3.2.3. Term Vectorization/Embeddings

The processed explicit and implicit features are added to the feature term embedding array $D$ as shown in Equation (10).

$$D = [K, T, S, A, J, I] \tag{10}$$

Here, $K$, $T$, $S$, $A$, $J$, and $I$ represent the processed data on the keywords, title, abstract, authors names, journal name, and institution name, respectively.

Then, $D$ is converted into a term vectorization using Word2Vec, forming the initialized feature matrix for the subsequent CNN model. Word2Vec is a flat neural network model that maps terms into a multidimensional numerical space, where the position in the numerical space indicates the semantic information of the term [48]. To compute the term embeddings, we used Skip-gram, a Word2Vec term vector model method, to predict the occurrence probability of contextual environment terms based on the central term. The use of pre-trained term vectors greatly improves the classification performance of CNN models [49]. Consistent with Timoshenko and Hauser [50], in this study, we set the sliding window size $c$ to 5 and the term vector dimension $d$ to 20 to define the parameters of the Skip-gram Word2Vec model. The array $D^*$ is the input for the term vector model, and the output is a term vector matrix $D^* \in i^{d \times n}$, which is used as the input for the CNN model.

### 3.3. Deep Learning for Literature Classification

Compared to traditional machine learning algorithms, deep learning models have performed better in classifying large-scale texts [42,51,52]. Deep learning models can learn high-level features at a deep level, starting from primary features at a superficial level by connecting neurons [26]. The specified input data are first fed to a feature extraction network, and the extracted features are then fed to a classification network. For the term vector matrix $D^*$ built in this study, the deep-learning model CNN can learn both the aggregated features and the detailed features contained in the various scientometric information. An important advantage of CNNs is that they have the ability to automatically extract features [2]. A typical convolutional layer usually uses a pooling function in its last phase to change the output of the layer. In this study, we used max-pooling [30], one of the widely used pooling functions that returns the maximum output within a rectangular neighborhood. Max-pooling is a pooling operation that computes the maximum value for patches of a feature map and uses it to create a down-sampled (pooled) feature map. The result is down-sampled or pooled feature maps that highlight the most present feature in the field, rather than the average presence of the feature as in the case of average pooling. In practice, this method has been shown to perform better than average pooling in automated feature extraction, such as feature extraction in text mining applications [37,38].

As shown in Figure 5, the CNN model consists of an input layer, a convolution layer, a pooling layer, and a SoftMax layer, and uses a gradient descent method to inversely adjust the weighting parameters [30]. The input layer of the CNN is the term vector matrix $D^*$, and the convolution layer performs the convolution operation on the original feature matrix with multiple convolution kernels to form the feature map. The feature map is then pooled to reduce the dimensionality and set the threshold. The pooling layer filters out the unusable features and keeps the essential features. Finally, the SoftMax layer converts the sampled output vectors from the pooling layer into probability values for the literature content and method, using a fully connected SoftMax function to predict article categories.

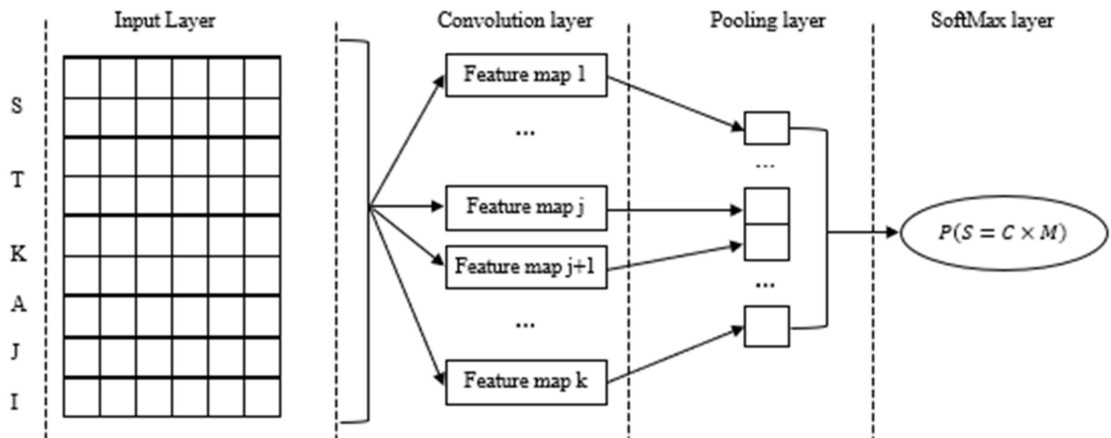

**Figure 5.** Structure of the CNN model.

For a collection of research content set *S*, both research content and research method are used as the output of the CNN model, and the resulting combination is modeled as in Equation (11).

$$S = C \times M \tag{11}$$

where *C* denotes the research content labels set, *M* denotes the research method labels set, and *S* denotes the combination of research content and research method labels, thereby achieving dual-label classification. The specific process is shown in Figure 6. For example, assuming that there are 4 research methods and 8 types of research content in a certain field, then 32 topic labels need to be defined, namely topic label 1, topic label 2, and up to topic label 32. If a paper is labeled with "Topic Label 32", it means that its research content and research method are "Research Content 8" and "Research Method 4" respectively.

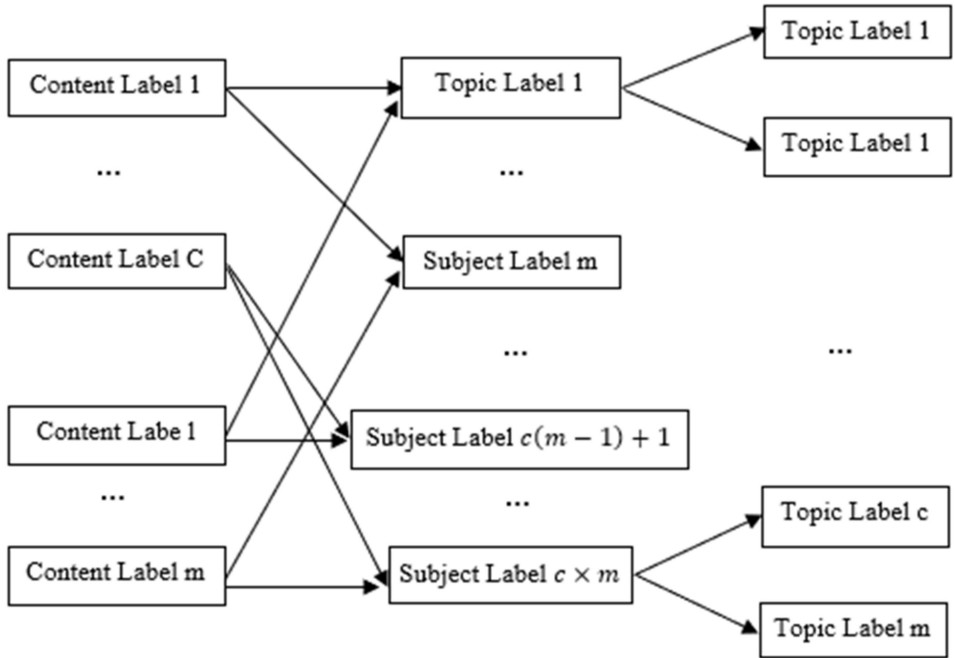

**Figure 6.** Process of implementing dual-label classification of literature.

## 4. Experimental Validation

To validate the utility and effectiveness of the above classification model, this study analyzes research content and method from the data science and analytics literature. Data science and analytics span multiple domains and disciplines; therefore, there is no general

agreement in the literature on their scope, dimensions, and the characteristics of their components and interactions [53–57]. Salazar-Reyna et al. [30] described data mining, machine learning, and software tools as three overlapping subsystems of data science along two dimensions (i.e., experimental versus theoretical; descriptive versus prescriptive). Purnomo et al. [27] developed a schematic representation of the major domains that comprise data science and analytics (i.e., domain knowledge, statistics/mathematics, and computer science) and their overlap (i.e., machine learning, biostatistics, and data science). The literature is replete with publications on the use of data science, data analytics, and machine learning algorithms in various application areas such as healthcare, business, decision making, and marketing. However, it is unclear to what extent researchers in this field collaborate to share new insights and important results [30]. Therefore, a systematic approach to fine-grained classification of the currently published literature in the field of data science and analytics would be helpful to categorize these scholarly publications in the field of data science and analytics into granular subcategories and to establish future directions for the development and refinement of this area of research.

### 4.1. Data Source and Collection

The preliminary protocol for the search design included the use of five individual search terms (data science, data analytics, big data analytics, machine learning, and business analytics), three platforms (Scopus, ProQuest, and EBSCOhost), the use of Boolean operational rules (AND/OR), searching across all domains, and the primary exclusion criterion—publication in English-language journals. This search approach was repeatedly tested and modified to obtain a final set of appropriate guidelines for the study setting. To increase the exploratory nature of the search, we added synonyms (e.g., data mining, big datasets, and business intelligence) and techniques (e.g., data processing, text mining, machine learning, neural networks, and deep learning) to the original search terms, as well as the term data science (given the lack of uniformity between data analytics and data science in publications and academic texts) with the Boolean operator OR to the original search terms. To increase sensitivity, the Boolean operator was implemented in the abstracts rather than in all data elements or in the entire text content, which helps to control the scope.

The search covered the period from 2000 to 2022, and a total of 8874 records containing titles, study institutes, journals, keywords, abstracts, and other information were retrieved. After removing duplicates, processing errors, and missing values, 7647 records were finally sampled for annotation. The 7647 articles contain 13,977 key terms, and the most frequently occurring term is data analytics, with a total of 136 occurrences. A total of 785 journals are included, with the highest frequency for a single journal being 291 (*International Journal of Data Science and Analytics*—Springer) and 560 journals occurring more than twice. A total of 6785 research institutions are included, with the highest publication frequency for a single research institution being 82 (Boston University School of Information Management) and 1899 occurring more than twice. A total of 10,568 authors are recorded, with the highest frequency for a single author being 33 and 2262 authors occurring more than twice. The distribution of high-frequency terms, journals, and research institutions is shown in Table 3.

**Table 3.** Example of a journal—research content probability distribution.

|  | Research Content-1 | Research Content-2 | Research Content-3 | Research Content-4 | Research Content-5 |
|---|---|---|---|---|---|
| Institution-1 | 0.60 | 0 | 0 | 0.40 | 0 |
| Institution-2 | 0.15 | 0.10 | 0.75 | 0 | 0 |
| Institution-3 | 0 | 0 | 0.34 | 0.33 | 0.33 |
| Institution-4 | 0 | 1 | 0 | 0 | 0 |
| Institution-5 | 0.25 | 0 | 0.20 | 0.25 | 0.30 |

*4.2. Manual Annotation*

We identified 22 classification features and divided them into two categories: research content and research method. This was achieved through a preliminary literature review based on the classification method described by Luo et al. [58]. As shown in Table 4, these 22 labels describe the main research areas of data science and analysis. In terms of research content labels, "self-service analytics" refers to niche areas of data science and analytics research, such as augmented data analytics and self-service business analytics [59,60]. This study classifies research that uses mainly qualitative research methods to analyze concepts or interpretive strategies as theoretical research in terms of research methodology. Research that mainly uses econometric or quantitative methods to examine cross-sectional or time-series data at the macro and micro levels is classified as empirical research. Research that focuses on organizational behavior by constructing models and using specific data to analyze specific cases is classified as case studies. Systematic literature review (SLR) refers to applying data science and analytical techniques in various application areas, such as healthcare, marketing, and decision-making [30].

**Table 4.** Frequency distribution of high-frequency terms, journals, and research institutions.

| Subject Terms | Frequency | Journal | Frequency | Research Institution | Frequency |
|---|---|---|---|---|---|
| Data analytics | 136 | *International Journal of Data Science and Analytics* | 291 | Boston University | 82 |
| Big data analytics | 131 | *Data Science Journal* | 290 | California Institute of Technology | 77 |
| Machine learning | 109 | *International Journal of Data Science and Analytics* | 233 | Case Western Reserve University | 75 |
| Deep learning | 77 | *Intelligent Data Analysis* | 179 | Cornell University | 68 |
| Business analytics | 73 | *International Journal of Behavioral Analytics* | 166 | Davidson College | 66 |
| Business intelligence | 72 | *MIS Quarterly* | 133 | University of Chicago | 65 |
| Deep learning and neural networks | 68 | *Data Science and Management* | 115 | University of Georgia | 57 |
| Data mining | 68 | *Intelligent Data Analysis* | 115 | University of Michigan | 52 |
| Artificial intelligence | 67 | *The Journal of Finance and Data Science* | 106 | University of Notre Dame | 46 |
| Internet of Things | 63 | *Statistical Analysis and Data Mining* | 85 | University of Pennsylvania | 44 |

Considering the highly specialized nature of scientific literature and the uniformity of citation standards [32,61], we used experts for manual citation annotation rather than the popular crowdsourcing model. Citation and annotation of data under the crowdsourcing model is often performed by many non-experts, which improves citation efficiency but is not suitable for citing highly specialized scientific literature. As for the specific operations, we mainly use subject matter experts to identify a set of feature terms for each research content label (see Table 5) and combine the position and frequency of occurrence of the feature terms for research content labels. When more than two types of research content occur in an article, the research content labels are sorted by the frequency of the feature terms, and the research content labels with a high feature terms' frequency are selected.

**Table 5.** Topic labels of the literature.

| Categories | Topic Labels | No. |
|---|---|---|
| Research Content | Machine learning; business analytics; business intelligence; decision support systems; Internet of Things; big data analytics; deep learning and neural networks; data visualization, financial analytics; marketing analytics; data mining, text analytics, sentiment analytics, artificial intelligence, predictive analytics, operation research; prescriptive analytics; self-service analytics. | 18 |
| Research Method | Theoretical studies; empirical studies; case studies; systematic literature review (SLR) | 4 |
| Total | | 22 |

To ensure the accuracy of the annotation results, annotations were made independently by two master's students in data science and analytics research according to the annotation rules above. If the annotation results of two or more coders matched, the label category of the article was determined. The intercoder reliability and agreement between the two coders was tested using Krippendorff's alpha threshold as a measure of evaluator quality [62]. In general, a Krippendorff's alpha ($\alpha$) of 0.80 is considered an acceptable level of intercoder reliability and agreement [63]. However, when the annotation results of the two coders did not match, the article and intercoder agreement were reviewed by the subject matter experts. The final intercoder reliability in this process, as measured by Krippendorff's alpha coefficient, was 83%, suggesting an acceptable level of intercoder agreement.

*4.3. Experimental Analysis*

4.3.1. Evaluation Criteria

The classification of the research areas in the literature was evaluated using precision, recall, and $F1$ measure. The method used in this study is to consider the class labels to be evaluated as individual positive classes and the other classes as negative classes and to construct a confusion matrix for each class label. The number of samples correctly assigned to a class label is $TP$, the number of samples incorrectly assigned to that label is $FP$, the number of samples correctly assigned to other class labels is $TN$, and the number of samples incorrectly assigned to other class labels is $FN$. The values $P$, $R$, and $F1$ are obtained from Equations (12)–(14).

$$P = \frac{TP}{TP + FP} \tag{12}$$

$$R = \frac{TP}{TP + FN} \tag{13}$$

$$F1 = \frac{2 \times P \times R}{P + R} \tag{14}$$

4.3.2. Comparative Analysis

The data science and analytic literature classification results are shown in Tables 6–8. Table 6 shows the classification accuracy of our model for the topic categories of data science and analytic literature. Table 7 shows the classification accuracy of our proposed model based on different initial feature constructs. Table 8 shows the classification accuracy of our model compared to other machine learning benchmark models.

**Table 6.** Annotation features of disciplinary categories in data science and analytics.

| Research Content Label | Main Feature Terms |
|---|---|
| Machine learning | Database, computer vision, supervised learning, unsupervised learning, Reinforcement learning, neural network, classification, clustering, association rule mining. |
| Business analytics | Business, decision support systems, statistical model, descriptive analytics, diagnostic analytics, predictive analytics, prescriptive analytics, quantitative methods. |
| Business intelligence | Database, data warehouse, visualization, descriptive analytics, business performance management, key performance indicators, dashboard, scorecards, decision support. |
| Decision support systems | Online reviews, pricing research, consumer preferences. |
| Big data analytics | Recommendation algorithms, cloud computing. |
| Deep learning and neural networks | Deep learning, neural networks, long short-term memory. |
| Data visualization | Visualization techniques, graphics, descriptive analytics, data representation, communication, decision support. |
| Internet of Things | IoT data analytics, cloud computing, real-time streaming, network, smart manufacturing, interconnected devices, cloud manufacturing, fog computing, smart city. |
| Text analytics | Natural language processing text classification, topic modeling, social media, document frequency, corpus, lexicon, online reviews. |
| Sentiment analytics | Machine learning, user-generated content, opinion mining, voice, users, customers, subjective information, computational linguistics, biometrics, social network analysis. |
| Predictive analytics | Machine learning, predictive model, statistical analysis, supervised learning, unsupervised learning, reinforcement learning, classification, feature selection. |
| Artificial intelligence | Machine learning, augmented analytics, robotics, self-service analytics, deep learning, neural networks, decision making. |
| Operations research | Problem solving, optimization, decision making, prescriptive analytics, management science, simulation, supply chain management, planning, enterprise resource planning, risk management. |
| Prescriptive analytics | Management science, business performance management, optimization, decision making, sensitivity analysis. |
| Data mining | Statistics and modeling techniques, clickstream data. |
| Self-service analytics | Business user, report, dashboard, data-driven organizations, citizen data scientist, ad hoc analysis, queries, reports. |
| Financial analytics | Ad hoc analysis, forecast, business questions, financial data, financial risk. |
| Marketing analytics | Marketing campaigns, customer analytics, marketing channels, customer behavior, online reviews, brand management. |

**Table 7.** Classification results of data science and analytics literature topic categories.

| Research Topics | | Performance Indicators | | |
|---|---|---|---|---|
| **Category** | **Label** | **P** | **R** | **F1-Score** |
| Research Content | Machine learning | 0.95 | 0.95 | 0.95 |
| | Business analytics | 0.93 | 0.92 | 0.92 |
| | Business intelligence | 0.91 | 0.94 | 0.92 |
| | Decision support systems | 0.84 | 0.84 | 0.84 |
| | Big data analytics | 0.85 | 0.81 | 0.83 |
| | Deep learning and neural networks | 0.88 | 0.82 | 0.85 |
| | Data visualization | 0.87 | 0.91 | 0.89 |
| | Internet of Things | 0.59 | 0.56 | 0.57 |
| | Text analytics | 0.94 | 0.93 | 0.93 |
| | Sentiment analytics | 0.93 | 0.89 | 0.91 |
| | Predictive analytics | 0.88 | 0.84 | 0.86 |
| | Artificial intelligence | 0.88 | 0.88 | 0.88 |
| | Operations research | 0.88 | 0.86 | 0.87 |
| | Prescriptive analytics | 0.88 | 0.88 | 0.88 |
| | Data mining | 0.91 | 0.92 | 0.91 |
| | Self-service analytics | 0.89 | 0.85 | 0.87 |
| | Financial analytics | 0.74 | 0.74 | 0.74 |
| | Marketing analytics | 0.74 | 0.77 | 0.75 |
| Research Method | Theoretical research | 0.72 | 0.76 | 0.74 |
| | Empirical research | 0.93 | 0.91 | 0.92 |
| | Qualitative research | 0.91 | 0.89 | 0.90 |
| | Case study | 0.61 | 0.66 | 0.63 |
| | Systematic literature review | 0.72 | 0.79 | 0.75 |

**Table 8.** Comparison of data science and analytics literature topic classification results with different data inputs and preprocessing.

| Input Data and Preprocessing | Performance Indicators | | | | | |
|---|---|---|---|---|---|---|
| | Research Content | | | Research Method | | |
| | **P** | **R** | **F1** | **P** | **R** | **F1** |
| Classification model of this study | 0.73 | 0.74 | 0.73 | 0.88 | 0.84 | 0.86 |
| By directly adding the journal name, author, and institution | 0.62 | 0.61 | 0.61 | 0.74 | 0.71 | 0.72 |
| Title and abstract only | 0.72 | 0.73 | 0.72 | 0.76 | 0.75 | 0.75 |
| Using only English Snowball stop term list [43] | 0.70 | 0.71 | 0.70 | 0.76 | 0.77 | 0.76 |

As shown in Table 6, among the classification results of data science and analytics literature, machine learning has the highest *F*1 value of 95%, while the Internet of Things has the lowest *F*1 value of 57%. The *F*1 values of Financial Analytics and Marketing Analytics are relatively low, while the *F*1 values of the other research topics and categories are above 80%. The analysis shows that the research content with poor classification results has a relatively wide range of literature studies. For example, the research content in the literature labeled as the Internet of Things and related to data science and analytics has a

lower coverage rate. The inconsistency of the research content leads to a wide dispersion of literature features in the categories of the Internet of Things, and the classification results are relatively poor. For research method, except for the case study category, which has a low $F1$ value, the $F1$ value is above 85% for all other categories, and the classification results are superior. We found that the proportion of literature using case studies in data science and analytics is relatively small for the case study category, accounting for only 7.36% of the total labeled literature. The small amount of literature may lead to poor feature extraction and overfitting of the model for the case study research method.

We conduct comparative experiments using different methods to investigate the usefulness of the original feature matrix created in this study. The experiments are based on the research method proposed in this study, in which only one feature term is changed at a time while the other feature terms remain unchanged. The results are shown in Table 7. The accuracy rate of the title information-based literature classification model for research content classification is 72%, the recall rate is 73%, and the macro F1 value is 74%. The accuracy rate for research method classification is 88%, the recall rate is 80%, and the macro $F1$ value is 81%. The macro F1 values for research content and research method decreased by 9% and 11%, respectively, by directly including the original data of authors, institutions, and journal titles in the feature matrix. When using the CNN algorithm to classify documents, which was applied by other researchers who used only article titles and abstract data [9], the macro $F1$ values for research content and research method were 2% and 3%, respectively. When the lexicon of domain features is not included in the data preprocessing and the original data is classified by Porter's English Snowball stop term list only, the macro $F1$ values for research content and research method differ by 4%. This is clear evidence that mapping author, institution name, and journal name features and creating an initial feature matrix help improve the model's classification performance.

To test the effectiveness of the CNN algorithm for fine-grained classification of scientific literature, we used common machine learning algorithms as benchmark experiments, including Naïve Bayes classifier, support vector machines (SVM), and k-nearest neighbor (KNN) [42,64,65]. All feature terms were used identically in the experiments, but the models were different. The empirical results in Table 9 show that the best classification results are obtained with the classification model of this study (based on the CNN algorithm). The NB algorithm performs better in document topic classification than the traditional machine learning algorithms, but the gap is more obvious than the CNN algorithm. The macro $F1$ values of the research content classification results differ by 9% and the research method differs by 13%.

**Table 9.** Classification results of literature topics in data science and analytics using different models.

| Model | Performance Indicators | | | | | |
|---|---|---|---|---|---|---|
| | Research Content | | | Research Method | | |
| | **P** | **R** | **F$_1$** | **P** | **R** | **F$_1$** |
| Fine-grained classification model based on CNN | 0.72 | 0.73 | 0.74 | 0.88 | 0.80 | 0.81 |
| SVM | 0.57 | 0.60 | 0.58 | 0.69 | 0.41 | 0.51 |
| NBM | 0.64 | 0.67 | 0.65 | 0.70 | 0.67 | 0.68 |
| KNN | 0.50 | 0.50 | 0.50 | 0.69 | 0.45 | 0.54 |

Overall, the comparative analysis in Tables 7 and 8 shows that the method proposed in this study improves the macro $F1$ values of literature classification results. This proves the effectiveness of the method proposed in this study in addressing the problem of fine-grained classification of scientific literature.

### 4.3.3. Limitations and Future Research

Thematic classification of published literature based on scientometric information also presents several challenges.

First, caution is needed when generalizing the results of this work because each research method has different biases, such as database bias (resulting from the use of a limited database) and interpretation bias (resulting from the use of multiple researchers' interpretations of the content of publications). To reduce the impact of these biases in this study, the authors used multiple platforms to collect relevant publications (ProQuest, EBSCOhost, and Scopus). Each of these platforms has access to different databases. Since the data used in the experiments of this paper are relatively small, further large-scale literature classification experiments should be conducted to validate the proposed method's effectiveness further.

Second, due to the complexity of the research topics and the multidisciplinary nature of the literature, it is not easy to make a coherent classification of the scientific literature. In this study, we only classify the topics based on the most frequently occurring terms, which may lead to an inaccurate classification. For example, in the literature on data science and analytics, we named machine learning, predictive analytics, and business analytics based on the frequency of the feature terms; however, the classification result of the model is machine learning, which shows the inadequacy of the output in a single category. In further research, the output layer of the CNN model should be improved to develop a multi-category output and increase the accuracy of literature topic classification.

Third, in deep learning, the feature extraction and modeling steps are performed automatically after data training. Future research could therefore be devoted to improving the proposed method by incorporating feature selection techniques such as principal component analysis (PCA), regularization, matching pursuit, random projection, random forests, and LASSO.

Finally, another problem is that the classifications of the literature are partially based on the manual coding of subject matter experts who have determined the labels for the sub-categories; this imposes certain subjective limitations. Further research should consider a combination of machine learning algorithms and expert knowledge to identify research topics in the literature and improve the model's ability to classify the literature automatically.

## 5. Conclusions

Advancement and development in a particular area of research are illustrated by the ever-growing body of scientific literature. This accumulation of literature describes various developments and innovations that occur over time and contains potentially valuable information that can be evaluated and classified to explain current emerging trends. In this study, we present a scientometric classification and analysis model for scientific literature based on convolutional neural networks (CNNs). Three dimensions, namely publication features, author features, and content features, were divided into explicit and implicit features to form a set of scientometric terms through explicit feature extraction and implicit feature mapping. First, we filter this scientometric information that characterizes the literature topic and build a lexicon of features and stop terms based on all scientometric titles, key terms, and abstracts in the training set. We then extract the explicit features such as keywords, titles, and abstracts from the scientometric information and map the implicit features such as authors, journal titles, and institutions to create a feature map. The feature map is then trained with weighted term vectors and used as input to the CNN model. Finally, the CNN model is used to perform a dual-label classification based on the published literature and articles' research content and methods.

The development of such an updated classification scheme is illustrated by a case study from the data science and analytics literature. The empirical results show that the macro F1 values of the proposed model for the two categories of research content and research method are 0.74 and 0.81, respectively. Compared to traditional machine learning methods, the classification results are significantly better and outperform the results obtained using

only explicit features. Building a multi-faceted and fine-grained classification framework based on this approach provides the opportunity to explore interactions between disciplines (i.e., inter- and intra-disciplinary knowledge flows) in more detail, and allows for more detailed identification of inconsistencies between different classification systems. By using a minimal set of textual data, the approach presented in this study can be practically generalized to other datasets (e.g., artificial intelligence and big data analytics literature). Additional bibliographic metadata would likely improve the overall performance of the classification method. An interesting avenue would be to use full text, which provides more textual data and better accuracy of explicit features mapping and transformation. Additionally, it would be very promising to study and analyze in detail the classification ambiguities resulting from the algorithms' predictions.

Drawing on the findings and limitations of the current study, we believe that future research should focus on intensifying theoretical and applied research in this area in four directions: Assessing the costs and benefits of data science and analytics applications, conducting normative analytics studies, using data science and analytics to analyze the decision-making process, and validating the proposed research methods in a short period of time in different contexts using different databases and publishing platforms. Therefore, we believe that our work will stimulate similar future studies to explore the limits of machine learning and data science classification capabilities.

**Author Contributions:** Conceptualization, M.D.; methodology, M.D. and S.A.; software, S.A.; validation, M.D. and S.A.; formal analysis, M.D.; investigation, M.D.; resources, M.D. and W.M.; data curation, M.D.; writing—original draft preparation, M.D.; writing—review and editing, W.M. and L.A.; visualization, W.M.; supervision, W.M.; project administration, M.D., W.M. and L.A. All authors have read and agreed to the published version of the manuscript.

**Funding:** This research received no external funding.

**Conflicts of Interest:** The authors declare no conflict of interest.

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
