# Peer review of "Scientometric Analysis and Classification of Research Using Convolutional Neural Networks: A Case Study in Data Science and Analytics"

_electronics, doi:10.3390/electronics11132066_

Round 1

Reviewer 1 Report

The authors develop a deep learning approach for scient metric analysis and classification of scientific literature based on convolutional neural networks. By setting dominant and recessive features as input to CNN model, achieve a two-label classification of literature based on research content and methods. Then, the author compares the model with similar machine learning classification methods and finds that its parameters are better than other models, which proves the effectiveness of the model. Finally, the author gives the defects of the model and the future research direction. Overall, the manuscript has enough merit to be published in the journal with some minor modification as stated below,

  1. The introduction is too concise, it is best to further explain the importance of literature classification for researchers to manage literature.
  2. Could you further explain the basis for selecting explicit features and implicit features? The description in this section is a little brief.
  3. Line 57, I admit to not being familiar with citations 12,13, but is it appropriate to cite them here? Please briefly explain.
  4. It is recommended to move lines 56-67 to after line 136.
  5. For Figure 1-6, it is recommended to replace a clearer and more intuitive picture.
  6. Line 162, The key terms you mentioned are in the feature term lexicon. Could you further elaborate on the feature term lexicon?
  7. Line 165, You mentioned "the English snowball stop term list". Could you compare your new term list with it?
  8. In equations (4) to (6), a format misaligned error occurred.
  9. Line 213, “exceeds a certain threshold” in matching threshold. But how to define this threshold?
  10. Line 331, “we added synonyms (e.g., data analytics, data science, large data sets, and business intelligence)”. The word “data analytics” and “data science” has already appeared in line 326, obviously not synonymous.
  11. Line 373-375, “If the annotation results of two or more coders matched, the label category of the article was determined; however, if the annotation results of all three coders did not match, the article was referred to the subject matter experts.”. How to handle if the annotation results of two coders did not match?
  12. The advantage of deep learning is that it can automatically extract features. This paper also needs to set features to train the model. Could you make an improvement on this point?

Author Response

Point 1: The introduction is too concise, it is best to further explain the importance of literature classification for researchers to manage literature.

Response: Thank you for your comment on this important point. In the revised manuscript, we have updated the "Introduction" section and added text to discuss the importance of literature analysis and classification for researchers and platform management. Please note the highlighted text in the "Introduction" section.

We have also added text to highlight the main contribution of this work and specify the new aspects of the study. We have also made clear why we used Deep Learning and CNN techniques to solve the research problem.

Point 2: Could you further explain the basis for selecting explicit features and implicit features? The description in this section is a little brief

Response: Thank you for raising this important point. In the revised manuscript, we have added text to explain the basis for selecting explicit and implicit features. Explicit and implicit features differ significantly in their identification methods; explicit features can be easily identified by their direct association with the classification label. In contrast, there are no obvious linguistic and syntactic clues for identifying implicit features; instead, they must be determined based on their deep semantic features and are usually those features that are not directly associated with the classification label. Therefore, identifying implicit features is still one of the most difficult tasks in scientometric analysis of literature. A very effective and useful method to extract implicit features is to use association rules to identify the implicit feature by finding the association of a certain feature with the classification labels; in this study, the classification labels are presented as research contents and research methods.

Please refer to the highlighted text (lines 168-183 in 3. Model development).

Point 3: Line 57, I admit to not being familiar with citations 12,13, but is it appropriate to cite them here? Please briefly explain.

Response: Thank you for pointing this out. In the revised manuscript, we have deleted these two references as citing them there is not appropriate.

Point 4: It is recommended to move lines 56-67 to after line 136.

Response: Thank you for highlighting this important point. In the revised manuscript, we have deleted these lines and moved them to the suggested section. Please note the highlighted text in section "3. Model development".

Point 5: For Figure 1-6, it is recommended to replace a clearer and more intuitive picture.

Response: Thank you for your comment on this important point. In the revised manuscript, we have updated the figures to make them clearer and more intuitive.

Point 6: Line 162, The key terms you mentioned are in the feature term lexicon. Could you further elaborate on the feature term lexicon?

Response: Thank you for your comment on this important point. In the revised manuscript, we have added text to explain the feature term lexicon and how it was created in this study. Please refer to the highlighted text in the “3.1 Building stop and feature terms lexicons” section.

Point 7: Line 165, You mentioned "the English snowball stop term list". Could you compare your new term list with it?

Response: Thank you for your comment on this important point. In the revised manuscript, we have updated the text to explain the difference between our dictionary and the English snowball stop term list. In addition to Porter's English snowball list, the stopword lexicon in this study includes sentence-initial terms and terms with low frequency and unclear categorical features. Compared to Porter's English snowball list, our stopword lexicon consists of frequently mentioned terms related to open source tools (e.g., KNIME, RapidMiner, Weka, MS Power BI), programming languages, and libraries (e.g., Python, R Script, STATA, SQL, and NLP).

Please refer to the highlighted text in the “3.1 Building stop and feature terms lexicons” section.

Point 8: In equations (4) to (6), a format misaligned error occurred.

Response: Thank you for your comment on this important point. Equations were revised and corrected.

Point 9. Line 213, “exceeds a certain threshold” in matching threshold. But how to define this threshold?

Response: Thank you for raising this point. In the revised manuscript, we have updated the text to explain how we specified the threshold for calculating probabilities. The threshold value is a hyperparameter, and after experimentation, it is assumed that the threshold value is set to 0.7, i.e., if an author's  value is greater than 0.7, the author is assigned to that research method label, otherwise, the placeholder 0 is used.

Please refer to the highlighted text in the “3.2.2 Implicit feature mapping” section.

Point 10. Line 331, “we added synonyms (e.g., data analytics, data science, large data sets, and business intelligence)”. The word “data analytics” and “data science” has already appeared in line 326, obviously not synonymous.

Response: Thank you for raising this point. In the revised manuscript, we have updated the text to correct these errors. Please refer to the highlighted text in the “4.1 Data Source and Collection” section.

Point 11. Line 373-375, “If the annotation results of two or more coders matched, the label category of the article was determined; however, if the annotation results of all three coders did not match, the article was referred to the subject matter experts.”. How to handle if the annotation results of two coders did not match?

Response: Thank you for raising this point. In the revised manuscript, we have added new text to explain how we tested intercoder reliability and agreement between the two coders. Please refer to the highlighted text in the “4.2 Manual annotation” section.

Point 12. The advantage of deep learning is that it can automatically extract features. This paper also needs to set features to train the model. Could you make an improvement on this point?

Response: Thank you for raising this point. In the revised manuscript, we have updated the text to further explain the automatic feature extraction method. In this study, we used Max-Pooling for automatic feature extraction. Max-pooling is a pooling operation that computes the maximum value for patches of a feature map and uses it to create a down-sampled (pooled) feature map.

Please refer to the highlighted text in the “3.3 Deep Learning for Literature Classification” section.

Reviewer 2 Report

The paper focuses on classifying research papers according to their content and methodological approach using deep learning.

The topic is interesting and worth investigating. 

1. The diagram in Figure 1 is hard to understand. It is not very clear to which part of the diagram the texts on the left belong.

2. It is not clear why the authors have not chosen to use a corpus specific stopwords list, constructed automatically.

3. It is not very clear how the association between the authors and their probability table is created in section 3.2.2. The authors are kindly asked to better explain if this a manual or an automatic process. 

4. The same observation can be made for journals and research institutions. For example the paper states that "After that, the probability values of the label's research content and research method were calculated for each research institution". The paper should more clearly specify how this has been achieved.

5. In section 3.2.2 it is not very clear if O represents the name of  the institution. The authors are kindly asked to clarify this issue.

6. Additionally, is J the actual journal title, as specified in the paper or a probability?

7. It is also unclear why the authors have chosen to use dual-label classification, instead of training two separate models. 

Author Response

Point 1. The diagram in Figure 1 is hard to understand. It is not very clear to which part of the diagram the texts on the left belong.

Response: Thank you for your comment on this important point. In the revised manuscript, we have updated the figures to make them clearer and more intuitive.

Point 2. It is not clear why the authors have not chosen to use a corpus specific stopwords list, constructed automatically.

Response: Thank you for your comment on this important point. In the revised manuscript, we have updated the text to explain the difference between our dictionary and the English snowball stop term list. In addition to Porter's English snowball list, the stopword lexicon in this study includes sentence-initial terms and terms with low frequency and unclear categorical features. Compared to Porter's English snowball list, our stopword lexicon consists of frequently mentioned terms related to open source tools (e.g., KNIME, RapidMiner, Weka, MS Power BI), programming languages, and libraries (e.g., Python, R Script, STATA, SQL, and NLP).

Please refer to the highlighted text in the “3.1 Building stop and feature terms lexicons” section.

Point 3. It is not very clear how the association between the authors and their probability table is created in section 3.2.2. The authors are kindly asked to better explain if this a manual or an automatic process.

Response: Thank you for raising this point. In the revised manuscript, we have updated the text to explain how we specified the threshold for calculating probabilities. The threshold value is a hyperparameter, and after experimentation, it is assumed that the threshold value is set to 0.7, i.e., if an author's  value is greater than 0.7, the author is assigned to that research method label, otherwise, the placeholder 0 is used.

Please refer to the highlighted text in the “3.2.2 Implicit feature mapping” section.

Point 4. The same observation can be made for journals and research institutions. For example the paper states that "After that, the probability values of the label's research content and research method were calculated for each research institution". The paper should more clearly specify how this has been achieved.

Response: Thank you for raising this point. In the revised manuscript, we have updated the text to explain how we specified the threshold for calculating probabilities. Please refer to the highlighted text in the “3.2.2 Implicit feature mapping” section.

Point 5. In section 3.2.2 it is not very clear if O represents the name of the institution. The authors are kindly asked to clarify this issue.

Response: Thank you for pointing this out. We have corrected this typo in the revised manuscript. We have detected O and replaced it with J to refer to the journal. Please refer to the highlighted text in the “3.2.3 Term Vectorization/Embeddings” section.

Point 6. Additionally, is J the actual journal title, as specified in the paper or a probability? Response: Thank you for pointing this out. We have corrected this typo in the revised manuscript. J stands for the name of the journal, not for a probability. Please refer to the highlighted text in the “3.2.3 Term Vectorization/Embeddings” section.

Point 7. It is also unclear why the authors have chosen to use dual-label classification, instead of training two separate models.

Response: Thank you for pointing this out. In the revised manuscript, we have updated the text to explain that we address the dual-label classification problem in the context of scientometric classification of literature, which requires assigning labels to each document for research content and methods with high descriptive accuracy. The task of dual-label classification is to train a function to predict the unknown sample and return two target labels for each of the documents. Unlike traditional classification, dual-label classification represents a sample by an eigenvector and two target labels, instead of just one exclusive label. The traditional classification approach consists of training different classification prediction labels separately. This approach is characterized by low training and testing efficiency and reasonable memory consumption when the set of target labels is quite large.

Please refer to the highlighted text in the “3. Model Development” section.

Round 2

Reviewer 2 Report

I would like to thank the authors for the changes made.